# Concurrent Drought and Temperature Stress in Rice—A Possible Result of the Predicted Climate Change: Effects on Yield Attributes, Eating Characteristics, and Health Promoting Compounds

**DOI:** 10.3390/ijerph16061043

**Published:** 2019-03-22

**Authors:** Alphonsine Mukamuhirwa, Helena Persson Hovmalm, Hans Bolinsson, Rodomiro Ortiz, Obedi Nyamangyoku, Eva Johansson

**Affiliations:** 1Department of Plant Breeding, Swedish University of Agricultural Sciences, P.O. Box 101, 23053 Alnarp, Sweden; helena.persson@slu.se (H.P.H.); rodomiro.ortiz@slu.se (R.O.); eva.johansson@slu.se (E.J.); 2Department of Food Technology, Engineering and Nutrition, Lund University, P.O. Box 124, 22100 Lund, Sweden; hans.bolinsson@food.lth.se; 3College of Agriculture, Animal Sciences and Veterinary Medicine, University of Rwanda, P.O. Box 210 Ruhengeri, Rwanda; obedingyoku@gmail.com

**Keywords:** *Oryza sativa*, grain yield, quality, bioactive compounds, drought, temperature

## Abstract

Despite the likely increasing co-occurrence of drought and heat stress, not least in equatorial regions, due to climate change, little is known about the combinational effect of these stresses on rice productivity and quality. This study evaluated the impact of simultaneous drought and temperature stress on growth, grain yield, and quality characteristics of seven rice cultivars from Rwanda, grown in climate chambers. Two temperature ranges—23/26 °C night/day and 27/30 °C night/day—together with single or repeated drought treatments, were applied during various plant developmental stages. Plant development and yield were highly influenced by drought, while genotype impacted the quality characteristics. The combination of a high temperature with drought at the seedling and tillering stages resulted in zero panicles for all evaluated cultivars. The cultivar ‘Intsindagirabigega’ was most tolerant to drought, while ‘Zong geng’ was the most sensitive. A “stress memory” was recorded for ‘Mpembuke’ and ‘Ndamirabahinzi’, and these cultivars also had a high content of bioactive compounds, while ’Jyambere’ showed a high total protein content. Thus, climate change may severely impact rice production. The exploitation of genetic diversity to breed novel rice cultivars that combine drought and heat stress tolerance with high nutritional values is a must to maintain food security.

## 1. Introduction

Rice (*Oryza sativa* L.) is a major staple food crop in the world which contributes to both food security and income generation, particularly within developing countries [1]. Rice has a semi-aquatic phylogenetic origin and thrives in diverse environments if provided with sufficient water [2] and temperatures ranging from 20 to 35 °C. Around 50% of the rice cultivated area is irrigated, whereas rainfed cultivation covers 43% and submerged cultivation covers 7% [2]. In Rwanda, rice growing areas include lowland valleys at around 900 m above sea level with temperature over the growing season ranging from 17 to 33 °C and highland valleys at around 1700 m above sea level with temperatures from 15 to 27 °C [3]. According to Meteo Rwanda measurements, humidity is normally relatively low in Rwanda rice cultivation areas (50–75%). The predicted global warming constitutes a serious threat to rice production as well as to the quality of the produced rice. Temperature stress and drought are predicted to increase to a higher extent in tropical and subtropical regions [4], which are the main rice producing areas. The predicted temperature increase for Rwanda is 0.9–2.2 °C until the mid-21st century, with increasing number of extreme weather events [5].

High temperature stress (above the optimum of 25–30 °C [6]) or drought conditions have negative effects on plant development, including irreversible injury to growth and development of the plant, reduction of photosynthesis [7], reduction in number of panicles per plant and peduncle elongation, limitations of pollen production [4], no swelling of pollen grains, and increased spikelet sterility [8]. Low temperatures (below the optimum of 25–30 °C [6]) lead to inhibited seedling growth, reduced panicle development, delayed heading, poor panicle exertion, low spikelet fertility, and poor grain quality [9]. Besides affecting growth and grain yield, water and temperature stresses change the quality and chemical composition of rice. Levels of starch, phenols, flavonoids, and phytic acid were shown to decrease, whereas antioxidant capacity, content of amylose, oxalic acid, calcium, and iron seemed to increase in drought-stressed grains [10]. In another study, gelatinization temperature decreased in drought stressed rice grains, whereas it increased under heat stress [11]. Therefore, the predicted increase in drought and heat stresses constitute a great threat to rice productivity and quality characteristics, and it may affect lives of millions of the world’s population, especially in poor areas of the tropical and subtropical regions [4], where rice constitute a staple food. Consequent to reduction in grain yield, food insecurity, malnutrition, extreme poverty, and other health and social problems will increase if no appropriate measures are taken to mitigate the effects of climate change on food production.

Adaptation measures such as the modification of planting date, alternate wetting and drying, mulching, aerobic rice cultivation [12], selection of cultivars with stress escape/avoidance possibilities [13], and breeding for drought or heat/cold stress are different solutions that have been adopted to mitigate rice productivity at climate change. For example, the International Rice Research Institute (IRRI) has developed drought tolerant cultivars such as ‘Sahbhagi Dhan,’ ‘Sahod Ulan,’ and ‘Sookha Dhan’ grown today in India, the Philippines, and Nepal, respectively [13]. However, multiple stresses may have contrasting effects, and simultaneous occurrence may cause interaction effects that cannot be extrapolated from the effects of individual stress [14]. Treatment strategies need to be, therefore, carefully adopted to fit the circumstances. The co-occurrence of drought and heat stress extremes is likely to increase—both in intensity and frequency—with the actual global warming. Simultaneously, cold nights can still be an issue in rice growing highland valleys of, for instance, Rwanda. Gaps still exist in the knowledge of concomitant environmental effects on rice growth, productivity, and the quality of harvested grains. Hence, understanding rice responses to a combination of temperature and drought stress, as well as their recurrence across plant developmental stages, is needed.

The aim of this study was to investigate how drought and temperature stress combinations during different growth stages affect rice growth, grain yield, eating quality, and health promoting compounds. The acquired knowledge will form the basis for the successful cultivation and breeding of novel rice cultivars that will sustain rice production and nutritional quality within a changing climate, thereby contributing towards human health by reducing poverty in densely populated rice growing areas. 

## 2. Materials and Methods

### 2.1. Plant Material and Cultivation

#### Plant Material

Seven rice cultivars (Table 1) obtained from the Rwanda Agricultural Board (RAB) were used in this study. The cultivars were released in 2010 by the Agricultural Research Institute of Rwanda (ISAR), currently known as RAB, except ‘Zong geng,’ which was introduced from China in the 1960s, and ‘Intsindagirabigega,’ which was introduced in 2002 from WARDA (currently Africa Rice Centre) and released in 2004. The cultivars were selected based on their medium water requirements, i.e., approximately 909 L of water are needed to produce 1 kg of rough rice [15]. The other main characteristics of the used cultivars [16] are presented in Table 1.

### 2.2. Growth Chamber

For the present experiment, five replications of a total of seven cultivars were grown using seven different drought treatments and two different temperature treatments. To control the effect of environmental factors, except the selected drought and temperature treatments, the rice cultivars were grown in growth chambers in the biotron of the Swedish University of Agricultural Sciences (SLU) at Alnarp. SLU biotron chambers are known for their high accuracy and low level of between-chamber effects [17]. To avoid the effects of endogenous rhythm in plants [18], all plants in the experiment were sown at the same time. Thus, two growth chambers were used in this experiment: The first chamber was set at a day and night temperature of 26/23 °C (low temperature, T1), whereas the second was set at 30/27 °C (high temperature, T2) for the whole growing period. The selection of temperatures for the experiment were based on Kohl [19] when growing rice in a controlled environment (23–31 °C), the optimum temperature (25–30 °C) in the field according to Wopereis et al. [6], and on the known field conditions of Rwanda [3]. Moreover, we considered the predicted climate change of a 2 °C increase [2] and low night temperatures being present in Rwanda [3], both of which are known to be harmful to rice growth and grain yield [20]. Thus, we used the low optimum temperature of 25 °C as a starting point for the calculation of night temperatures. By decreasing and increasing this optimum temperature by 2 °C, we set the night temperatures to 23 °C and 27 °C, respectively. Thereby, the selected night temperatures can be seen as representatives of an increase in night temperature from climate change in highland and lowland conditions of Rwanda, respectively. Day temperatures were set to 3 °C higher than night temperatures following the procedure by Harrington [21]. Thereby, we ended up with day and night temperatures of 26/23 °C (low temperature, T1) and 30/27 °C (high temperature, T2), representing climate predicted climate changes in the lowland and highland of Rwanda, respectively. For both chambers, a light intensity of 350PAR µmol s^−1^ was used [22] with 11 h of light and 13 h of darkness, whereas the atmospheric relative humidity was 70% [23], the latter also representing the field conditions for rice cultivation in Rwanda.

### 2.3. Sowing and Watering

Pots with the size 6 × 10 × 12 cm were filled with soil and placed in big plastic trays. Two seeds per pot were sown, and, after emergence, the most vigorous seedling in each pot was left to grow. A total of 35 plants (five plants per cultivar) were grown in each tray and seven trays, i.e., a total of 245 plants, were kept in each growth chamber. The soil was kept wet by regular watering from soil surface up to three weeks after sowing. Thereafter, water was added to the plastic trays, and plants got access to the water through holes in the bottom of the pots. Water depth was maintained at 5 cm in the tray by regularly adding water to it. 

### 2.4. Drought Treatments

The drought treatments applied in the present study followed methodology developed at IRRI [23] and also applied at Africa Rice in Tanzania. In this study, the aim was to evaluate the effects of drought at different growth stages, and, therefore, a number of different drought treatments were applied, i.e., well-watered throughout the growth cycle (D0), drought during the seedling stage only (DS), drought during the seedling and active tillering stages (DST), drought at the active tillering stage only (DT), drought at the active tillering and reproductive stages (DTR), drought at reproductive stage only (DR), and drought at every stage, i.e., the seedling, active tillering, and reproductive stages (DSTR). To initiate drought stress, the trays were drained from water using a tiny tube, and water was not added to the trays again until the drought treatment was finalized. For drought during the seedling stage (DS), the plants were drought treated for two weeks. For drought during the active tillering stage (DT), water was withheld for two weeks; plants were then watered from the soil surface for one day and drought stressed again for another two weeks. For drought at the reproductive stage (DR), the drought stress cycle was one week of drought, one day of watering, and one week of drought. The combined drought treatments (DST, DTR, and DSTR) used the same treatments as for the separate treatments but in combined doses. Soil water content was measured in the pots at the end of each drought treatment period using an AT Delta—T device (Theta probe type HH1). The probe readings were between 36 and 87 in drought stressed pots, depending on drought treatment and cultivar; in non-stressed pots, it was 513, i.e., a soil moisture content of 7–17% in drought treated pots. After each drought treatment described above, watering was resumed by adding water to the tray.

### 2.5. Data Recording in the Growth Chambers

After each drought treatment, the number of tillers plant^−1^ was counted. Drought sensitivity was evaluated applying the 0–9 scoring scale for leaf rolling and drying, as proposed by the IRRI [24], although scoring was carried out separately at the seedling and tillering stages. The recovery rate was calculated 10 days after resuming watering, as the percentage of recovered plants over the total number of plants that were present before the drought treatment. Days to flowering were counted as days after sowing to 50% of plants flowering. Number of panicles plant^−1^, panicle length, number of spikelets panicle^−1^, and spikelet sterility rate (percentage of sterile grains over total number of grains panicle^−1^) were counted at harvesting time. Each mature plant was harvested individually. The panicles were manually threshed, and the grain yield plant^−1^ was measured using a precision balance.

### 2.6. Analysis of Nutritional Quality

#### 2.6.1. Sample Preparation

In order to obtain large enough samples for milling and nutritional quality analyses, rice grains from each of the cultivar × drought × temperature treatment were pooled, and each of these samples was thereafter stored individually at −20 °C. A working sample was thereafter collected from each of these stored samples for freeze-drying in 48 h. The freeze-dried whole grains were thereafter grinded using an IKA-WERKE grinder type A10, (SKAFTE medlab, Mindelheim, Germany). The flour was stored at −20 °C until further analyses. All analyses of nutritional quality were then made in triplicate from the flour.

#### 2.6.2. Total Protein Content

The total nitrogen content was determined following the nitrogen combustion method [25] using a nitrogen/carbon analyzer (Flash 2000NC Analyzer, Thermo Scientific, Waltham, MA, USA). The total protein content was calculated by multiplying the total nitrogen content by a conversion factor of 5.7 [26]. 

#### 2.6.3. Total Phenolic Content and Total Antioxidant Capacity

The total phenolic content (TPC) was determined according to Singleton et al. [27], and the total antioxidant capacity (TAC) was analyzed following Pérez-Jiménez’s and Saura-Callixto’s [28] methods with small modifications, as described in Mukamuhirwa et al. [29]. 

#### 2.6.4. Amylose Content

The amylose content (AmC) was determined following the spectrophotometric iodine analysis of starch proposed by Hofvander et al. [30], except that the defatting of rice starch was carried out by the use of 95% ethanol [31].

#### 2.6.5. Gel Consistency

The gel consistency (GC) of each sample was measured following the gel consistency test for eating quality of rice developed by Cagampang et al. [32]. Briefly, 0.2 mL of 95% ethanol containing 0.025% thymol blue was added to 100 ± 0.5 mg of rice flour in a culture tube. After vortexing, 2 mL of 0.2N KOH were added, and the mixture was vortexed again. The tubes were covered by concave marble glass and directly put in a boiling water bath (95 °C) for 8 min. The tubes were then cooled at room temperature for 5 min and placed on ice water for 15 min. After cooling, the tubes were horizontally placed on a millimetric paper, and the length of the gel was measured from the bottom of the tube to the top of the gel run.

#### 2.6.6. Gelatinization Temperature

The gelatinization temperature (GT) was determined using a differential scanning calorimetry (DSC, Seiko 6200) method following Gunaratne et al. [33] with small modifications. About 3 mg of rice flour was added into a coated aluminum DSC pan (TA Instruments) in duplicate, and 10 µL of water were gently added to the flour sample. After water addition, the pans were quickly hermetically sealed to avoid water evaporation. For measurement, the scanning rate was set to 10 000 degrees min between 20–150 °C. An empty pan served as reference.

### 2.7. Statistical Analysis

Simple regression analyses were carried out to understand the proportion of the contribution of drought treatment, temperature regimes, and cultivars on the variation in number of tillers per plant^−1^, leaf rolling, and leaf drying, similarly as described in Malik et al. [34]. Drought treatments and cultivars were analyzed according to the mean values for each character evaluated by the regression analyses. An analysis of variance (ANOVA), followed by calculations of means separated with Tukey’s test (*p* < 0.05), was carried out using the Statistical Analysis Systems (SAS), Cary, North Carolina [35]. A principal component analysis (SAS program) was applied to understand variations and relationships among various parameters and treatments.

## 3. Results

### 3.1. Early Plant Growth—Drought Sensitivity Score, Number of Tillers Plant^−1^ and Time to Flowering

The applied drought treatment explained more than 40% of the variation in leaf rolling and drying at both the seedling and tillering stages, while the cultivar was the major determinant of the number of tillers plant^−1^ (Table 2). The combination of drought and cultivar explained more than 45% of the variation in leaf rolling and drying at the seedling and tillering stages (Table 2). There was no correlation between soil water content in the stressed pots at the end of the drought treatment and leaf rolling or leaf drying.

A principal component analysis (Figure 1) clearly showed the drought at the tillering stage as being the main component determining leaf rolling and drying at the tillering stage, i.e., DST, DT, DTR, and DSTR (encircled by a dotted line) are treatments that have positive values for the second principal component (Figure 1b), as is shown also for leaf rolling and drying at the tillering stage (Figure 1a). Similarly, positive values of the first principal component were found both for samples with drought treatment at the seedling stage (DS, DST, and DSTR; encircled by a full line, Figure 1b), and for leaf rolling and drying at the seedling stage (Figure 1a). The first principal component explained 48% of the variation, while the second principal component explained 32% of the variation. Among the cultivars, ‘Zong geng’ (No 7) was found to be the most sensitive to drought stress, showing high values of the first principal component (high values of leaf rolling and drying at the seedling stage) during seedling drought and high values of the second principal component (high values of leaf rolling and drying at tillering) during drought at the tillering stage. The number of tillers plant^−1^ decreased, while leaf rolling and leaf drying increased after drought stress at the seedling and tillering stages (Figure 1a).

There were no significant interactions between cultivar and drought or between cultivar and temperature regarding recovery rate and flowering time. The recovery rate at the seedling stage was significantly decreased for plants being drought treated at the seedling stage (DS, DST, DSTR; Table 3). Similarly, the recovery rate at the tillering stage was significantly decreased after drought at the seedling stage, followed by drought at the tillering stage (Table 3). Combined drought at the seedling and tillering stages (DST, DSTR) resulted in a significant delay in flowering (39 days) as compared to plants being not drought treated (D0; Table 3). High temperature treated plants flowered significantly earlier (30 days) than low temperature treated plants (Table 3). 

### 3.2. Rice Production—Grain Yield and Yield Components

The means of grain yield and yield components for all cultivars and treatments of this study are presented in Appendix A. Generally, the applied drought treatments heavily affected the presence and number of panicles plant^−1^ and, thereby, grain yield (Table 4). The D0 treatment (no drought) at a high temperature resulted in the significantly highest number of panicles plant^−1^. Combined drought treatment at the seedling and tillering stages (DST, DSTR) at a high temperature had a detrimental effect on the presence of panicles plant^−1^ and, thereby, yield, resulting in zero panicles plant^−1^ for all evaluated cultivars (Table 4). The cultivar ‘Intsindagirabigega’ showed a high number of plants with panicles, being able to better withstand all the evaluated drought treatments except the combined seedling and tillering treatments (DST and DSTR) at a high temperature (Table 4). The cultivar ‘Zong geng’ clearly showed the significantly lowest number of panicles plant^−1^, with a decrease at all drought treatments at a high temperature and at all drought treatments except DS and DST at the lower temperature (Table 4). The cultivars ‘Mpembuke’ and ‘Ndamirabahinzi’ both showed sensitivity to drought stress at the seedling stage (DS) at a low temperature, although being able to produce panicles at combined drought stresses at the seedling stage and at later stages (DST, DSTR). The numbers of spikelets panicle−^1^ and spikelets sterility were generally higher at a low temperature than at a high temperature (data not shown). Furthermore, all drought treatments resulted in severe decreases in yield as compared to when the plants were subjected to no drought (Figure 2). In drought stressed pots, a negative correlation (r = −0.28; *p* = 0.025) was found between soil and water content at the end of the experiment and grain yield.

### 3.3. Eating Quality and Health Promoting Compounds

The means of grain AmC, GC, GT, total protein content, TPC, and TAC for all cultivars and treatments are presented in Appendix A. In general, variation in the evaluated quality characteristics seemed to be more dependent on cultivar than on temperature or drought treatment. In a principal component analysis, the cultivars were shown to cluster into three groups (Figure 3b) corresponding to evaluated quality characteristics (Figure 3a). The first and second principal components explained 32% and 24% of the variation, respectively. The cultivars ‘Mpembuke’ (C4), ‘Ndamirabahinzi’ (C5), and ‘Nemeyubutaka’ (C6) showed positive values on the first principal component (encircled by a full line) independent of temperature and drought treatment (Figure 3b), indicating a generally high TAC and TPC content (Figure 3a) in these cultivars. Regression analyses also showed a high degree of cultivar determination for TAC (33.7%) and TPC (16.5%) content, and analysis of variance followed by mean comparisons confirmed that these cultivars have high TAC and TPC contents (results not shown). Furthermore, the cultivars ‘Ingwizabukungu’ (C1), ‘Intsindagirabigega’ (C2), and ‘Zong geng’ (C7) were clustered with negative principal component analyses values (encircled by a dashed line; Figure 3b) independent of temperature and drought treatment, indicating high AmC and GC content (Figure 3a), which was also verified by analyses of variance followed by mean comparisons (results not shown). The cultivar ‘Jyambere’ (C3), independent of treatments, formed the third cluster (encircled by a dotted line) with negative values on the second principal component (Figure 3b), indicating high protein and GT values (Figure 3a), which was verified by analyses of variance followed by mean comparisons (not shown). The principal component analysis indicates a possible influence of the growing temperature on GT values, with significantly higher GT values at the high temperature treatment (verified by analyses of variance and mean comparisons). In addition, regression analyses showed 10.4% of the variation in GT to be explained by temperature, as compared to 0.1% being determined by cultivar (results not shown).

## 4. Discussion

Our results clearly showed a negative concurrent effect of the temperature and single/repeated drought treatments at various growth stages on rice production and yield. Drought impacted to the highest extent the early plant development, while the genetic component—i.e., different cultivars— determined the quality of rice to the largest extent. Thus, our results clearly showed that the predicted increase in temperature and drought due to climate change will impact rice production, which, in turn, might influence food security in vulnerable highly populated areas around the equator.

A combination of a high temperature (27/30 °C) and recurrent drought at the seedling and tillering stages (which might be a predicted climate change scenario for lowland valleys of Rwanda) showed the most severe results for the plant development and yield, with no panicles plant^−1^ resulting in zero yield for all cultivars. However, for the rest of the combinations of drought and temperature treatments, the different cultivars reacted differently. Differences among cultivars towards drought conditions may be the result of a range of various characters in the plant, including both root and leaf characters [36]. Thus, cultivars capable of extracting more water from the soil and maintaining a good water status in the tissues are able to withstand drought and to produce grains. The cultivar ‘Zong geng’ showed the most severe reaction regarding panicles plant^−1^, resulting in a total absence after most of the treatments. This genotype is both taller and later flowering than the other cultivars when grown during field conditions (Table 1), which might explain part of its sensitivity to drought conditions, although additional studies are needed for a full understanding of variations in the reactions from the different genotypes. In addition, cultivar interactions with the soil water content and the time course of its’ changes may contribute to an understanding of drought treatments on various cultivars. The course of soil water content for each individual plants, cultivars, and treatments was not measured in the present study, and such interactions are an option for further studies. However, previous studies have indicated that leaf characteristics of rice, including leaf water status and leaf elongation rate, are limitedly sensitive to soil water content and deficits; instead, the root character was proposed as the main character related to drought sensitivity [37]. Furthermore, panicle exertion and abortion of secondary rachis branches has been reported as the major causes of yield reduction in rice under drought stress [38], which correspond to the results in the present study. The severe reduction in the number of panicles plant^−1^ and, consequently, reduction in grain yield in the present study indicate that a high temperature and recurrent drought at the seedling and tillering stages were severe for the rice plants. Drought conditions and the cultivation of rice in the present study were based on methodology developed at IRRI [23]. The high temperature combined with recurrent drought at the seedling and tillering stages might have resulted in more severe drought responses in rice than is normally found while rice is grown during field conditions. At present, rice is most likely not recommended to be grown at conditions with severe recurrent drought conditions at a high temperature. However, the severe drought responses from the rice in the present study might also be due to differences in growing conditions in the biotron compared to in field conditions. It is well known that biotron or green-house cultivations are not perfectly comparable to cultivation in field conditions. In the biotron, the rice plants were grown in pots, not allowing the roots to extract water deep in the soil; in addition, microbiota and other biotic factors are known to be different. Despite the fact that biotron conditions are not totally comparable to field conditions, our results predict a high risk of large reduction in rice yield in the predicted climate change scenarios. In previous studies, spikelet sterility has consistently been reported as an indicator of sensitivity to drought around flowering [39] and is directly associated with leaf water potential. However, in the present study, spikelet sterility was mainly found to be correlated to the temperature of rice cultivation and, to a lesser degree, to the drought treatments. The fact that spikelet sterility was not seen as correlated with drought treatments thereby verifies drought treatments as not being severe enough to result in differences in leaf water potential. Extreme temperatures have been found to affect spikelet fertility by causing poor anther dehiscence, pollen grains deficiency, and failure of pollen germination on stigmas [40,41]. Our results primarily showed high spikelet sterility at a low temperature, which has also been shown in other studies [42]. Spikelet sterility under the low temperature regime is a sign of this temperature being in the lower edge of the optimal range. However, we found comparable yields in high and low temperatures treated rice due to an increased spikelet fertility compensating for the low number of panicles plant^−1^ at a high temperature.

Early plant development was affected more by the drought treatments than by differences in temperature. The different drought treatments in the present study resulted in various effects on plant development. Thus, drought at the seedling and tillering stages resulted in leaf rolling and drying at these stages, and repeated drought treatments delayed the flowering—as did the low temperature treatment. A delayed flowering might be associated with reduced early growth. Previous studies have reported a reduction of relative water content from exposure to a low temperature [43], while drought resulted in a reduced synthesis of photosynthetic pigments [44]; both circumstances resulted in a delayed plant development. Early growth rate, leaf emergence, and tillering capacity was found to increase with rising temperatures, from 25 to 31 °C, as a result of increasing photosynthesis [45]. Thus, in our study, the shorter time to flowering in plants grown at a high temperature might be associated with high early growth rate. Drought at the seedling stage significantly reduced the recovery rate of the plants at both the seedling and tillering stages. Thus, this study showed the rice plants to be most sensitive to drought at the seedling stage, especially if followed with drought at the tillering stage and even more so at high temperatures. 

The cultivars included in this study reacted differently to the temperature and drought treatments. Effects of drought treatments on different genotypes is known to be the result of a range of various characters in the plant, including both root and leaf characters [36]. However, the only persistent difference recorded was that ‘Zong geng’ showed significantly higher drought sensitivity as compared to the other cultivars as it showed a higher leaf rolling and drying at both the seedling and tillering stages and the significantly largest decrease in number of panicles plant^−1^. Grown at a high temperature, ‘Zong geng’ produced spikelets only while not drought treated; grown at a low temperature, this cultivar produced spikelets only when well-watered and when treated with drought at the seedling stage (also combined with drought at the tillering stage). ‘Jyambere’, a cultivar reported in a previous study [29] as high yielding, did not show higher performance while drought treated than the other cultivars. Similarly to the present study, leaf rolling and drying were found to increase, while the tillering rate and number of panicles plant^−1^ decreased, with drought stress, although at significantly different extents in different cultivars [46].

The present study showed the limited influence of drought treatment on quality characteristics of rice, where only GT was clearly affected by and positively correlated with temperature. Previous studies have shown that an increased temperature reduces the grain starch and amylose content and also decreases the activity of granule-bound starch synthase [39], which might affect the GT. Independent of drought treatment, the cultivars ‘Mpembuke’ and ‘Ndamirabahinzi’ showed (together with ‘Nemeyubutaka’), high grain content of TAC and TPC, corresponding to previous results on these cultivars [29]. The results suggest that the content of phytochemicals/bioactive compounds might be under strong genetic control which means that it would be possible to breed for rice high in human health related compounds. High genetic diversity in bioactive compounds has been consistently found in rice cultivars [47,48]. Furthermore, QTLs with significant additive effects have been identified for some bioactive compounds in rice [49], which implies options to use marker assisted selection. In addition, AmC, GC, and protein content were found more related to cultivar than to treatment in the present study. Significant differences between genotypes for AmC and protein content have been observed in rice, whereas differences between drought and irrigation treatments [50] and between temperatures [51] have been negligible. The inheritance of AmC and GC in rice has been revealed to be predominantly governed by additive gene action [52,53]. Therefore, the genetic diversity present in the Rwandan rice cultivars offers opportunities to improve quality characteristics within the material.

Previous results have shown that exposure to stress at an early developmental stage can affect the tolerance and survival of plants when stressed again at a later stage [54]. Thus, plants seem to have a “stress memory” that allows them to cope with recurrent stress [55]. In the present study, the cultivars seemed to have a beneficial stress memory regarding drought stress to different degrees, and the presence of such a memory seemed to be influenced by the temperature treatment. The cultivars ‘Mpembuke’ and ‘Ndamirabahinzi’ seemed to have quite a beneficial strong stress memory, especially when grown at a low temperature, since they were not able to produce panicles and yield when drought stressed at the seedling stage but were able to produce panicles and yield after recurrent drought stresses. In addition, ‘Ndamirabahinzi’ seemed to have a stress memory if subjected to stress at a somewhat later stage, as a similar behavior was recorded for plants stressed at the tillering stage at first and then subjected to recurrent stresses. The sustained accumulation of stress signaling proteins or transcription factors at different levels from the initial state after stress alleviation was proposed as possible mechanism of plant memory [56]. Epigenetic changes involving DNA methylation or chromatin transformation without changing the nucleotide sequence have also been proposed [57] and may confer longer lasting effects than the metabolite accumulation. Further research may be able to elucidate the stress memory mechanisms under combined heat and drought.

The present study was carried out in climate chambers in the biotron at SLU, Alnarp. Despite the fact that climate chamber cultivations are not totally comparable with field cultivations, climate chamber experiments have the advantage of opportunities to control environmental effects on the cultivation. Plants grown in the same chamber have got exactly the same environmental conditions with the exception of treatments given to the plants (e.g., drought treatments in our case). The repeatability of experiments is also increasing with the use of climate chambers, as between-chamber effects are normally low, and, if they exist, they are mostly due to resolvable effects, i.e., the need for the reparation of malfunctioning components [17]. In our study, we used five replications of cultivars and treatments to secure statistically useful results as outcomes. However, a minor risk in our study is that the differences between temperature treatments we see might also partly be a result of between-chamber effects as used temperature treatments were divided in between chambers. Due to the high and well-known accuracy between the biotron chambers with a low level of between-chambers resolvable effects, we see this risk as low. In addition, the temperature results in this study correspond with results from field trials on drought stress for the same cultivars (unpublished results). In the present study, we also wanted to avoid experimental cultivation of the rice plants in various periods over the year; instead, we focused on at the same time planting to secure avoidance of effects of endogenous rhythms in plants [18]. Effects of such rhythms should absolutely have a bigger effect on the results than eventual small results between chamber unresolved effects [17,18]. 

Though the temperature sets were within the limits for rice development and productivity, this study showed that even small differences in temperature during rice growth may result in severe yield losses under recurrent drought. A small increase in temperature, which might come due to the predicted climate change, might be beneficial for rice production in highland areas of Rwanda, where the temperature currently is at the lower limit for rice production. However, this study shows that the positive effects of a small increase in temperature does not persist at recurrent drought conditions. Therefore, changes in temperatures prevailing in a specific environment should not be left behind when planning for improvement of rice for drought tolerance. There were large differences in panicle development between cultivars and between drought × temperature treatments. Hence, this trait may serve as selection criteria for rice tolerance to concurrent abiotic stresses.

## 5. Conclusions

The predicted climate change foresees a total increase in temperature combined with frequent dry spells. From this study, we can conclude that this scenario does not in general appear to be favorable for rice production. A moderate increase in temperature in “cool” regions, e.g., in highland valleys of Rwanda, combined with low or moderate drought stress, might result in a yield increase for rice. However, here, a temperature increase of only a few degrees, when combined with drought at the seedling and tillering stages, resulted in dramatic yield decreases for all cultivars. Some of the cultivars could not tolerate drought at all at the higher temperature. Rice is primarily cultivated in tropical and subtropical areas, being the staple food and major income crop for millions of poor people in densely populated areas. A severe decrease in production due to climate change will therefore heavily affect public health in these areas. This calls for a sincere screening to find rice capable of surviving an increased temperature with spells of drought. Here, cultivar differences in survival and production was clearly seen among the evaluated cultivars, although none of them a survived high temperature combined with drought at both the seedling and tillering stages. In this context, understanding the molecular background for some of the cultivars showing better ability to sustain the combined increase in temperature and dry spells is a necessity. Though drought and temperature combination did not affect the quality characteristics to a high extent, the combination of these stresses indirectly affect the nutritional quality and human health promoting compounds, since the quantity of produced grains becomes insufficient to meet the ideal daily intake and to fulfil the required nutritional content resulting in malnutrition. Thus, thorough screenings of cultivars that combine high quality and the capacity to withstand combined stresses at all stages of plants development and understanding involved mechanisms have to capture the attention of rice researchers.

## Figures and Tables

**Figure 1 ijerph-16-01043-f001:**
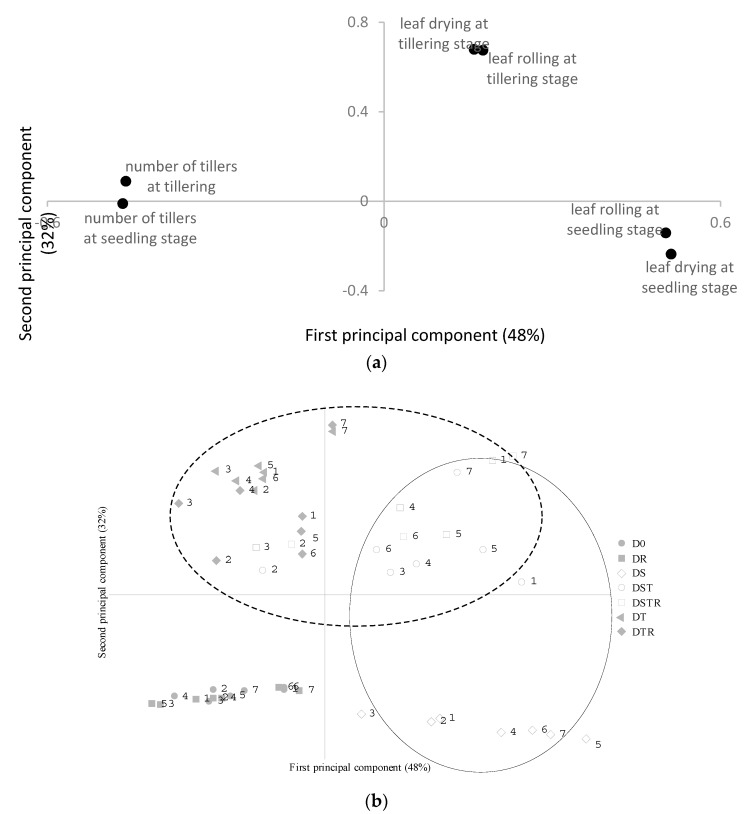
Loading (**a**) and score (**b**) plots from principal components analysis of leaf rolling, leaf drying, and number of tillers during drought at vegetative stages. Treatments, including drought at the seedling stage, are encircled by a full line, while treatments, including drought at tillering, are encircled by a dotted line. D0: Well-watered during the whole growing season; DS: Drought treatment at the seedling stage; DST: Drought treatment at the seedling and tillering stage; DT: Drought treatment at the tillering stage; DTR: Drought treatment at the tillering and reproductive stages; DR: Drought treatment at reproductive stage; DSTR: Drought treatment at the seedling, tillering, and reproductive stages. Numbers refer to cultivars as follow; 1: ‘Ingwizabukungu’; 2: ‘Intsindagirabigega’; 3: ‘Jyambere’; 4: ‘Mpembuke’; 5: ‘Ndamirabahinzi’; 6: ‘Nemeyubutaka’; and 7: ‘Zong geng’.

**Figure 2 ijerph-16-01043-f002:**
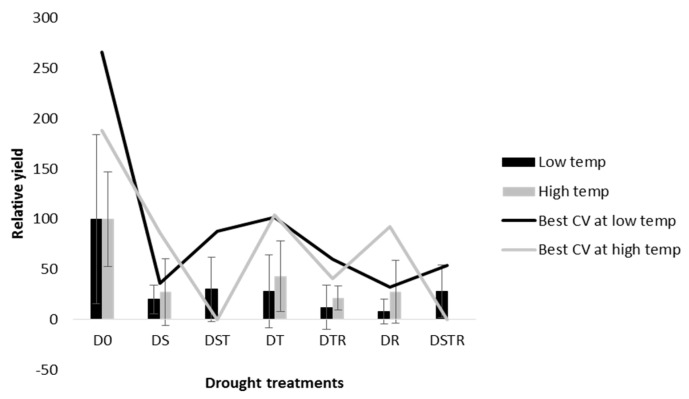
The relative yield (as compared to mean values of the cultivars at no drought treatment at temperatures applied, respectively) of all cultivars at each of the drought treatments at each of the temperatures (bars) and of the cultivar with the highest at each treatment and temperature (line). Error bars represent standard deviation of mean values for yield of all the cultivars.

**Figure 3 ijerph-16-01043-f003:**
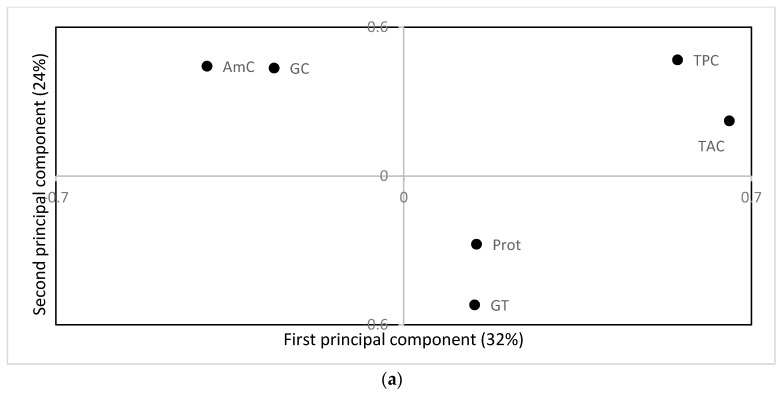
Loading (**a**) and score (**b**) plot from principal components analysis of amylose content (AmC), gel consistency (GC), gelatinisation temperature (GT), total antioxidant capacity (TAC), total phenolic content (TPC), and total protein content (Prot) in grains of rice cultivars stressed with drought at different development stages. Treatments corresponding to high TAC, TPC, and protein content are encircled by a full line, whereas a dashed line encircles the treatments corresponding to high or low eating and cooking qualities (AmC, GC, and GT), and a dotted line encircles the treatment corresponding to low content of all analyzed nutritional quality traits. T1: 26/23 °C; T2: 30/27 °C; DS: Drought treatment at seedling stage; DST: Drought treatment at the seedling and tillering stage; DT: Drought treatment at the tillering stage; DTR: Drought treatment at the tillering and reproductive stages; DR: Drought treatment at reproductive stage; DSTR: Drought treatment at the seedling, tillering, and reproductive stages. Numbers refer to cultivars as follow; C1: ‘Ingwizabukungu’; C2: ‘Intsindagirabigega’; C3: ‘Jyambere’; C4: ‘Mpembuke’; C5: ‘Ndamirabahinzi’; C6: ‘Nemeyubutaka’; and C7: ‘Zong geng’.

**Table 1 ijerph-16-01043-t001:** Characteristics of rice cultivars included in this study.

ID	Name/code	Popular Name	Characteristics						
			Type	Plant Height	Flag Leaf	Panicle Exertion	Tillering Ability	Lifespan (Days)	Potential Yield (t ha^−1^)
1	N/A	‘Ingwizabukungu’	*Indica*	Intermediate	Intermediate	N/A	N/A	N/A	N/A
2	WAT 1395-B-24-2	‘Intsindagirabigega’	*Indica*	Intermediate	Intermediate	Well exerted	Medium	120–150	8.0
3	WITA 4	‘Jyambere’	*Iindica*	Intermediate	Intermediate	Moderate well exerted	Medium	152	10.9
4	WAB923-B-6-AL1	‘Mpembuke’	*Iindica*	Intermediate	Intermediate	Moderate well exerted	Medium	170	8.3
5	WAB 569-35-1-1-1-HB	‘Ndamirabahinzi’	*Indica*	Intermediate	Intermediate	Well exerted	Medium	143	7.6
6	WAB 880-1-38-20-28-P1-HB	‘Nemeyubutaka’	*Indica*	Intermediate	Erect	Well exerted	Medium	152	9.3
7	Zong geng	‘Kigoli’	*Japonica*	Tall	Intermediate	Moderate well exerted	Medium	180	6.0

Source: ISAR, 2010. NA: information not available.

**Table 2 ijerph-16-01043-t002:** Percentage of explanation (obtained through the coefficient of determination (R^2^) from simple linear regression analysis) of temperature (T), drought (D), cultivar (C), and their combinations on numbers of tillers plant^−1^, leaf rolling, and leaf drying at the seedling (S) and tillering (Tl) stages.

	No of Tillers Plant^−1^		Leaf Rolling		Leaf Drying	
	S	Tl	S	Tl	S	Tl
T	0.49	0.78	0.07	1.2	1.21	2.22
D	7.71	6.88	65.9	43.6	70.40	59.60
C	7.71	13.6	1.0	1.4	1.64	2.16
TD	2.82	0.02	10.6	12.0	5.11	2.77
TC	2.82	0.30	0.38	2.1	0.29	0.72
DC	9.24	8.85	66.9	44.8	71.70	61.00
TDC	3.17	0.08	10.9	12.3	5.31	2.95

T: temperature; D: drought; C: cultivar; TD: temperature and drought; TC: temperature and cultivar; DC: drought and cultivar; TDC: temperature, drought and cultivar.

**Table 3 ijerph-16-01043-t003:** Recovery rate at the seeding (S) and tillering (Tl) stages and time to flowering.

Cultivar	Recovery Rate (%)		Time to Flowering (DAS)
	S	Tl	
‘Ingwizabukungu’	65.0 a	38.0 a	91.6 b
‘Intsindagirabigega’	92.5 a	68.0 a	113.1 a
‘Jyambere’	87.5 a	52.0 a	112.3 a
‘Mpembuke’	62.5 a	50.0 a	113.9 a
‘Ndamirabahinzi’	45.0 a	48.0 a	105.8 ab
‘Nemeyubutaka’	73.3 a	65.5 a	107.7 ab
‘Zong geng’	60.0 a	22.2 a	121.5 a
Temperature			
Low	75.0 a	54.86 a	120.5 a
High	64.3 a	44.57 a	90.3 b
Drought			
D0	95.7 a	87.1 a	94.8 b
DS	38.6 c	32.9 c	111.4 b
DST	64.3 b	35.7 c	133.7 a
DT	100.0 a	41.4 bc	105.8 b
DTR	98.6 a	45.7 bc	111.7 b
DR	95.7 a	77.1 ab	103.7 b
DSTR	80.0 ab	38.6 c	133.2 a

Means followed by the same letter within a column are not significantly different between cultivars or temperatures or droughts according to Turkey’s test at *p* ˂ 0.05D0: Plants were well-watered along the growing cycle. DS: Drought stress at the seedling stage; DST: Drought stress at the seedling and tillering stages; DT: Drought at the tillering stage; DTR: Drought at the tillering and reproductive stages; DSTR: Repeated drought at every developmental stage, S: seedling stage, Tl: tillering stage.

**Table 4 ijerph-16-01043-t004:** Presence (+) and absence (−) of panicles plant^−1^ (and thereby grain yield) and percentage of plants per rice cultivar and treatment without any panicle. Cultivars 1: Ingwizabukungu; 2: Intsindagirabigega; 3: Jyambere; 4: Mpembuke; 5 Ndamirabahinzi; 6: Nemeyubutaka; and 7: Zong geng were grown at a low or high temperature and with six different drought treatments, D0: No drought stress, DS: Drought at the seedling stage, DST: Drought at the seedling and tillering stages, DT: Drought at the tillering stage, DTR: Drought at the tillering and reproductive stages, DR: Drought at reproductive stage, DSTR: Drought at the seedling, tillering, and reproductive stages.

Treatment	Cultivar							Plants per Temp × Drought with No Panicles (%)
Temp Drought	1	2	3	4	5	6	7	
Low D0	+	+	-	+	+	+	+	37.1 ab
Low DS	+	+	+	-	-	+	+	80.0 cde
Low DST	+	+	+	+	+	+	+	57.1 bcd
Low DT	+	+	+	+	-	+	-	74.3 cde
Low DTR	+	+	+	-	+	-	-	74.3 cde
Low DR	+	+	-	-	+	-	-	85.7 de
Low DSTR	-	+	+	+	+	+	-	65.7 bcd
High D0	+	+	+	+	+	+	+	11.4 a
High DS	+	+	+	+	-	-	-	74.3 cde
High DST	-	-	-	-	-	-	-	100.0 e
High DT	+	+	+	+	+	+	-	54.3 bcd
High DTR	+	+	+	+	+	+	-	65.7 bcd
High DR	+	+	+	+	+	+	-	51.4 bc
High DSTR	-	-	-	-	-	-	-	100.0 e
**Plants per cultivar with no panicles (%)**	62.9 ab	50.0 a	61.4 ab	68.6 ab	72.9 bc	58.6 ab	91.4 c	

Means followed by the same letters within a column for treatment or within a row for cultivar are not significantly different at *p* < 0.05 according to Tukey’s mean comparison test.

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
