# Peer review of "Concurrent Drought and Temperature Stress in Rice—A Possible Result of the Predicted Climate Change: Effects on Yield Attributes, Eating Characteristics, and Health Promoting Compounds"

_ijerph, 2019, doi:10.3390/ijerph16061043_

Round 1
Reviewer 1 Report
The authors slightly improved the manuscript as compared to the second version.
Unfortunately, the authors don’t give more information about the time course of the soil water content that could explain differences between varieties and also reasoning of the severe yield reduction under drought (which might be not found in farmers fields).
Regarding the statistical treatment of the data, I would suggest to do only comparisons of the varieties within a certain growth conditions, respectively time point (for a statistical comparison between treatments, indeed, one would repeat the whole experiment at least three times). This might also help to focus the study more on the characterization of the rice varieties.
Author Response
The authors slightly improved the manuscript as compared to the second version.
Thank you for the positive comment
Unfortunately, the authors don’t give more information about the time course of the soil water content that could explain differences between varieties
We thank the reviewer for pointing out this important topic. Changes have been made within the manuscript in order to better describe the drought treatments and their relation to soil water content and to highlight the topic.
and also reasoning of the severe yield reduction under drought (which might be not found in farmers fields).
Thanks for pointing out that this issue is still not satisfactorily discussed. We have made amendments to the manuscript in order to clarify and make the discussion more thorough.
Regarding the statistical treatment of the data, I would suggest to do only comparisons of the varieties within a certain growth conditions, respectively time point (for a statistical comparison between treatments, indeed, one would repeat the whole experiment at least three times). This might also help to focus the study more on the characterization of the rice varieties.
Thanks for bringing up that these issues are not yet solved in the manuscript. We have now added clarifications and discussed issues of growth chamber experiments and possible influences of various factors including repetitions of experiments within and between chambers.
Reviewer 2 Report
All comments have been addressed.
Author Response
All comments have been addressed.
Thank for the positive feedback. We are thankful for the valuable comments and scientific critic of the reviewer towards the improvement of the quality of our manuscript.
We hope thereby that the manuscript is now publishable in IJERPH.
Round 2
Reviewer 1 Report
no further comments
This manuscript is a resubmission of an earlier submission. The following is a list of the peer review reports and author responses from that submission.
Round 1
Reviewer 1 Report
The authors considerably improved the manuscript as compared to the first version. In particular, the authors could give some clarifications concerning the chosen temperatures (also in the context of the climatic conditions in the different regions of Rwanda.
Moreover, the authors included information about the characterization of the drought stress, respectively, the determination of the soil water content. However, it would be appropriate to include the main outcome of the determination of the soil water content into the Result chapter (e.g. time course of soil water content). I wonder whether there are some correlations between the soil water content at the end of the drought cycle and growth characteristics (or even yield) of the plants.

Author Response
Open Review
(x) I would not like to sign my review report
( ) I would like to sign my review report
English language and style
( ) Extensive editing of English language and style required
( ) Moderate English changes required
(x) English language and style are fine/minor spell check required
( ) I don't feel qualified to judge about the English language and style
Yes | Can be improved | Must be improved | Not applicable | |
Does the introduction provide sufficient background and include all relevant references? | ( ) | (x) | ( ) | ( ) |
Is the research design appropriate? | ( ) | (x) | ( ) | ( ) |
Are the methods adequately described? | ( ) | (x) | ( ) | ( ) |
Are the results clearly presented? | ( ) | ( ) | (x) | ( ) |
Are the conclusions supported by the results? | ( ) | (x) | ( ) | ( ) |
Comments and Suggestions for Authors
The authors considerably improved the manuscript as compared to the first version. In particular, the authors could give some clarifications concerning the chosen temperatures (also in the context of the climatic conditions in the different regions of Rwanda.
Moreover, the authors included information about the characterization of the drought stress, respectively, the determination of the soil water content. However, it would be appropriate to include the main outcome of the determination of the soil water content into the Result chapter (e.g. time course of soil water content). As presented in the material and methods, the soil water content was determined after ending the drought treatment at seedling and tillering stages before resuming watering.
I wonder whether there are some correlations between the soil water content at the end of the drought cycle and growth characteristics (or even yield) of the plants. At the end of the drought treatment period, soil water content varied depending on the cultivar. For example, the soil in the pots containing’ Intsindagirabigega’ had low water content but low leaf drying and high yield while other cultivars such as ‘Nemeyubutaka’ had high soil water content , and low leaf drying and relatively low yield whereas pots containing ‘Zong geng’ had the highest water content but the highest leaf drying and the lowest yield. Overall, there was no correlation (r = -0.027; P = 0.83) between soil water content and leaf drying; however, a significant negative correlation (r = -0.28; P=0.025) was noted between soil water content and grain yield in drought stressed pots.

Reviewer 2 Report
In “Concurrent Drought and Temperature Stress in Rice—A Possible Result of the Predicted Climate Change: Effects on Yield Attributes, Eating Characteristics and Health Promoting Compounds”, the authors describe an experiment in which they test the effect of a projected increase in temperature in combination with drought stress on the characteristics of Rwandan rice. Specifically, the experiment includes exposing plants to drought stress at developmental phases: seedling, tillering, and reproduction, and in combination. The authors find that drought exposure during the seedling stage (both alone and in combination with other stages) has the greatest effect on plant developmental traits and yield across cultivars. Interestingly, cultivar rather than stress most greatly impacted components of grain quality. The authors conclude that predicted climate change might affect crop yields in the near future and management strategies need to be improved. The presentation of the results was very clear, and the conclusions that followed were logical.
Major comments:
From what I read, only one true replicate of the experiment was performed, and the analyses was based on pseudoreplication of plants within a single chamber per temperature. At least a second replicate is necessary. If multiple replicates were performed, this must be clarified in the methods.
Minor comments:
Introduction:
The bulk of this paper discusses drought effects on rice. However, the introduction does not discuss what is known about drought on rice—rather the focus is on heat stress. Please expand on the current literature on drought effects on rice.
Methods:
Line 92: please define what “medium” water requirements means.
Section 2.2: The description of temperature choice is somewhat repetitive. Consider rewording some of the description.
Section 2.5: Please add a description of Recovery Rate % found in Table 3.
Results:
Table 1:
Figures 1 and 3: Please add the missing labels for panels (a) and (b). Include add the % variance explained on the axis instead of in the legend.
Figure 2: In the figure legend, please describe what the errors bars are (e.g. standard error of the mean). Also, change “staples” to bars.
Table 3: What were the interactions between cultivar and drought for these traits? Please add these to the table and interpretation to the results.
Line 258: Remove the reference to stress memory.
Discussion:
Lines 395-404: I do not understand the description of stress memory. There seems to be a conflation between 1) exposure to stress at an early developmental stage that then has detrimental effects later and 2) exposure to stress at an early developmental stage that has a beneficial effect on later stress. Please clarify your definition of “stress memory” and its effect. The molecular description is good, but the phenotypic effects are unclear.
The following lines have grammar/spelling errors for correction:
Line 24: sedling à seedling
Line 35: in particular à particularly
Lines 46-47: amounts of spells of extreme weather conditions à number of extreme weather events
Line 48: missing “)”
Line 52: missing “)”
Lines 55-56: remove “were shown to”
Line 58: Thereby à Therefore
Lines 315-318: Please revise for clarity.
Line 328: missing “flowering” at end of line
Lines 333-336: Please revise for clarity.
Line 338: pikelet à spikelet
Lines 369-372: Please revise for clarity.
Author Response
Open Review
(x) I would not like to sign my review report
( ) I would like to sign my review report
English language and style
( ) Extensive editing of English language and style required
(x) Moderate English changes required
( ) English language and style are fine/minor spell check required
( ) I don't feel qualified to judge about the English language and style
Yes | Can be improved | Must be improved | Not applicable | |
Does the introduction provide sufficient background and include all relevant references? | ( ) | (x) | ( ) | ( ) |
Is the research design appropriate? | ( ) | ( ) | (x) | ( ) |
Are the methods adequately described? | ( ) | (x) | ( ) | ( ) |
Are the results clearly presented? | (x) | ( ) | ( ) | ( ) |
Are the conclusions supported by the results? | (x) | ( ) | ( ) | ( ) |
Comments and Suggestions for Authors
In “Concurrent Drought and Temperature Stress in Rice—A Possible Result of the Predicted Climate Change: Effects on Yield Attributes, Eating Characteristics and Health Promoting Compounds”, the authors describe an experiment in which they test the effect of a projected increase in temperature in combination with drought stress on the characteristics of Rwandan rice. Specifically, the experiment includes exposing plants to drought stress at developmental phases: seedling, tillering, and reproduction, and in combination. The authors find that drought exposure during the seedling stage (both alone and in combination with other stages) has the greatest effect on plant developmental traits and yield across cultivars. Interestingly, cultivar rather than stress most greatly impacted components of grain quality. The authors conclude that predicted climate change might affect crop yields in the near future and management strategies need to be improved. The presentation of the results was very clear, and the conclusions that followed were logical.
Major comments:
From what I read, only one true replicate of the experiment was performed, and the analyses was based on pseudoreplication of plants within a single chamber per temperature. At least a second replicate is necessary. If multiple replicates were performed, this must be clarified in the methods.
It is true that one chamber was used for each temperature treatment. Since we worked in a controlled environment (Biotron), we considered that replicate of growth chambers would have exactly the same conditions; therefore the replication of growth chamber could not be a source of variation. We rather emphasized on minimizing the experimental error by considering each pot within each treatment as a replicate. Analyses were then done on results obtained from five replicates per treatment.
Minor comments:
Introduction:
The bulk of this paper discusses drought effects on rice. However, the introduction does not discuss what is known about drought on rice—rather the focus is on heat stress. Please expand on the current literature on drought effects on rice. The drought effects on rice development and reproduction are presented together with the effects of high temperature because they produce more or less similar effects (lines 48–52), whereas the effects of drought on nutritional and chemical composition are also presented in lines 54–58.
Methods:
Line 92: please define what “medium” water requirements means. A cultivar described as having medium water requirement is a cultivar that uses around 909 liters of water to produce 1 kg of rough rice. Clarifications were made in the text.
Section 2.2: The description of temperature choice is somewhat repetitive. Consider rewording some of the description. Thank you for the remark. We revised the text
Section 2.5: Please add a description of Recovery Rate % found in Table 3.
The recovery rate was calculated, 10 days after resuming watering, as the percentage of recovered plants over the total number of plants that were present before the drought treatment. (also added in the text)
Results:
Table 1:
Figures 1 and 3: Please add the missing labels for panels (a) and (b). Include add the % variance explained on the axis instead of in the legend. Thanks for the comment. The percentage (%) variance explained was added to the axis labels and removed from the legend
Figure 2: In the figure legend, please describe what the errors bars are (e.g. standard error of the mean). Error bars represent standard deviation of mean values for yield of all the cultivars Also, change “staples” to bars. Thanks for the correction. Staples were replaced by bars in the text
Table 3: What were the interactions between cultivar and drought for these traits? Please add these to the table and interpretation to the results. Thanks for asking the question. However, there were no significant interactions between cultivar and drought for the recovery rate and time to flowering (we added this in the text).
Line 258: Remove the reference to stress memory. Thanks for the observation. The reference was removed in the text
Discussion:
Lines 395-404: I do not understand the description of stress memory. There seems to be a conflation between 1) exposure to stress at an early developmental stage that then has detrimental effects later and 2) exposure to stress at an early developmental stage that has a beneficial effect on later stress. Please clarify your definition of “stress memory” and its effect. The molecular description is good, but the phenotypic effects are unclear. Thanks for the comment. In the present study, exposure to early stress had beneficial effects on stress later at later stage especially with the cultivar ‘Mpembuke’ and ‘Ndamirabahinzi’. We clarified this in the text.
The following lines have grammar/spelling errors for correction:
Thanks for the corrections. All suggested corrections/revisions were considered in the text
Line 24: sedling à seedling
Line 35: in particular à particularly
Lines 46-47: amounts of spells of extreme weather conditions à number of extreme weather events
Line 48: missing “)”
Line 52: missing “)”
Lines 55-56: remove “were shown to”
Line 58: Thereby à Therefore
Lines 315-318: Please revise for clarity.
Line 328: missing “flowering” at end of line
Lines 333-336: Please revise for clarity.
Line 338: pikelet à spikelet
Lines 369-372: Please revise for clarity.

Round 2
Reviewer 2 Report
Because of the lack of replication in this study, I cannot support its publication as a complete study and not preliminary work. All other aspects were addressed.